# Electrospun Gelatin/Dextran Nanofibers from W/W Emulsions: Improving Probiotic Stability Under Thermal and Gastrointestinal Stress

**DOI:** 10.3390/foods14101725

**Published:** 2025-05-13

**Authors:** Yuehan Wu, Ziyou Yan, Shanshan Zhang, Shiyang Li, Ya Gong, Zhiming Gao

**Affiliations:** 1Glyn O. Phillips Hydrocolloid Research Centre, School of Life and Health Sciences, Hubei University of Technology, Nanli Road, Wuhan 430068, China; 2Cooperative Innovation Center of Industrial Fermentation (Ministry of Education & Hubei Province), Hubei University of Technology, Nanli Road, Wuhan 430068, China

**Keywords:** water-in-water emulsion, electrospinning, probiotic encapsulation, viability

## Abstract

Probiotics offer numerous health benefits; however, preserving their viability during processing and storage remains a major challenge. This study investigates the electrospinning of gelatin/dextran (GE/DEX) water-in-water (W/W) emulsions for *Lactobacillus plantarum* encapsulation. By varying dextran concentrations, the ways in which phase behavior, viscosity, and conductivity influence fiber formation and morphology were analyzed. Scanning and transmission electron microscopy confirmed core–shell nanofibers, while FT-IR revealed electrostatic interactions rather than chemical reactions between GE and DEX. Encapsulated probiotics exhibited enhanced viability under thermal stress (65 and 72 °C), storage (25 and 4 °C), and simulated gastrointestinal conditions, maintaining high viability (>8 log CFU/g) compared with free cells. Notably, gelatin-rich shell phases provided stronger protection, likely due to gelation properties restricting bacterial mobility. These findings demonstrate that electrospinning of W/W emulsions is an effective strategy to improve probiotic stability, offering potential applications in functional foods.

## 1. Introduction

Probiotics are ‘live microorganisms which, when administered in adequate amounts, confer a health benefit on the host’ according to the joint definition issued by the Food and Agriculture Organization of the United Nations (FAO) and the World Health Organization (WHO) [1]. Probiotic cells are exposed to a succession of hostile conditions, beginning with the heat, dehydration, oxidative stress, and osmotic shifts imposed during food processing and storage, and continuing with the low pH, digestive enzymes, bile salts, and dense resident microbiota they encounter along the gastrointestinal tract. Each stress can denature proteins, disrupt membranes, or otherwise compromise cell integrity, sharply reducing the number of live organisms that reach the intestine [2,3]. Hence, keeping probiotic cells alive throughout manufacturing, shelf-life, and passage through the gastrointestinal (GI) tract is difficult, as they are easily damaged by elevated temperature, low pH, and shifts in water activity [4,5]. As probiotic efficacy is directly linked to the number of surviving cells, preserving high viability under such conditions has become a major priority, prompting intensive research into encapsulation strategies that shield probiotics from environmental insults and thereby safeguard their functionality in both functional-food and pharmaceutical applications.

Current encapsulation strategies including spray drying [6,7], freeze drying [8,9], coacervation [10], and extrusion [11] have been employed to protect probiotics. Although these techniques can enhance survival to some extent, they frequently suffer from drawbacks, like thermal degradation, low encapsulation efficiency, and incomplete protection during GI transit. In contrast, electrospinning offers a promising alternative due to its ability to generate nanofibers with exceptionally high surface-to-volume ratios under relatively mild conditions [12,13]. The resultant fibrous mats often exhibit superior encapsulation efficiency, improved retention of bioactivity, and the potential for controlled release [14]. These advantages make electrospinning an appealing approach to addressing some of the limitations inherent in more conventional encapsulation processes.

Emulsion electrospinning, which produces core–shell structures or multi-phase fibers, is particularly advantageous for encapsulating sensitive bioactive substances [15,16,17]. Water-in-water (W/W) emulsions have gained increasing attention in this context because they eliminate the need for organic solvents [18,19,20,21]. Gelatin (GE) and dextran (DEX) exemplify such a polymer pair, forming distinct polymer-rich phases at certain concentrations [22,23]. Gelatin is prized for its gelation properties and biocompatibility [24], whereas dextran confers stability, favorable rheological characteristics, and additional charged sites that may influence electrospinning [25]. When the formulation crosses a critical composition window, the GE and DEX phases may exchange roles, converting the former continuous phase into droplets and vice versa. This switch alters the droplet diameter and rheology, thereby influencing the structural characteristics of the resulting fibers. Mastering this inversion behavior is essential for steering fiber formation and obtaining a uniform probiotic load in the polymer network [26,27,28].

In this study, the electrospinning of GE/DEX-based W/W emulsions to encapsulate *Lactobacillus plantarum* was systematically explored. *Lactobacillus plantarum* is a probiotic strain noted for its health-promoting properties, but it is vulnerable to environmental stress. The phase behavior, viscosity, conductivity, and morphological changes at varying polymer ratios, including conditions before and after phase inversion, were characterized. Subsequently, the heat tolerance and simulated GI survival of *Lactobacillus plantarum* embedded in the fibers were measured and compared with those of non-encapsulated cells. By demonstrating how polymer composition and phase inversion influence encapsulation efficiency and probiotic viability, this work provides valuable insights for designing improved probiotic delivery systems in functional foods.

## 2. Materials and Methods

### 2.1. Materials

Dextran (average molecular weight about 500 kDa) was supplied by Shanghai Macklin Biochemical Technology Co., Ltd. (Shanghai, China). Gelatin, dextran–fluorescein isothiocyanate (DEX-FITC), and a LIVE/DEAD^®^ BacLight^TM^ Bacterial Viability Kit were obtained from Sigma-Aldrich (Shanghai) Trading Co., Ltd. (Shanghai, China) Potassium bromide (KBr), sodium dihydrogen phosphate (NaH_2_PO_4_), ethanol, and bile salts were obtained from Sinopharm Chemical Reagent Co., Ltd. (Shanghai, China). Fluorescein isothiocyanate (FITC) was supplied by Shanghai Aladdin Biochemical Technology Co., Ltd. (Shanghai, China). The *Lactobacillus plantarum* subsp. plantarum CICC 6240 strain was supplied by China Industrial Microorganism Collection and Management Center. De Man–Rogosa–Sharpe (MRS) broth culture medium was provided from Qingdao Hi−Tech Park Haibo Biotechnology Co., Ltd. (Qingdao, China). Ultrapure water was employed in all experiments.

### 2.2. Incubation of Lactobacillus plantarum

*Lactobacillus plantarum* was first cultured in MRS broth at 30 °C for 24 h. After that, the bacterial cells were then collected by centrifugation (3000 rpm, 10 min, 4 °C). The sediment was subsequently washed with PBS and centrifuged, with this procedure being repeated three times in total. The cells were finally resuspended in sterile water to obtain a final concentration of 1 × 10^10^ colony forming units (CFU)/mL.

### 2.3. Preparation of Water-in-Water (W/W) Emulsions

Dextran and gelatin were separately dissolved in deionized water to prepare stock solutions of 40 wt% DEX and 20 wt% GE. These prepared solutions were mixed in specific proportions and stirred in a water bath at 40 °C for 3 min to prepare W/W emulsions with a fixed gelatin content of 10 wt% GE and varying DEX concentrations of 1.5 wt%, 3 wt%, 4.5 wt%, 6 wt%, 7.5 wt%, 9 wt%, 10.5 wt%, 12 wt%, 13.5 wt%, and 15 wt%, and named as 1.5% DEX, 3% DEX, 4.5% DEX, 6% DEX, 7.5% DEX, 9% DEX, 10.5% DEX, 12% DEX, 13.5% DEX, and 15% DEX, respectively. In the encapsulation process of probiotics, Lactobacillus plantarum was incorporated into the dextran phase, reaching a final concentration of 10^9^ CFU/mL in the W/W emulsion.

### 2.4. Preparation of Electrospun Fibers

A volume of 5 mL prepared W/W emulsion was drawn into a 10 mL syringe and secured on a LSP01-1A syringe pump (Qili Precision Pump Co, Ltd., Baoding, China) fitted with a 24G needle (inner diameter: 0.30 mm). Electrospun fibers were fabricated at a voltage of 16 kV through a DW-9303 high-voltage source (DongWen High Voltage Power Supply (Tianjin) Co., Ltd., Tianjin, China). The needle-to-collector distance was set at 15 cm and the flow rate was set of 1 mL/h, with an ambient temperature maintained at 60 °C. Throughout the experiment, the syringe pump and collector were placed inside a sealed acrylic enclosure. The temperature was maintained at 60 ± 2 °C using an infrared heating lamp, and air bubbles were eliminated from the needle before electrospinning.

### 2.5. Characterization

The droplet morphology of the emulsions was examined using microscopy. The droplet-size distribution was obtained from optical micrographs using ImageJ software (version 1.53k). The viscosity of the solutions was measured by employing a HAAKE RS6000 rheometer (Thermo Fisher Haake, Karlsruhe, Germany) equipped with a C60/1°Ti double cone/plate sensor (60 mm). Samples were pre-sheared at a fixed frequency of 0.1 1/s for 1 min, then subjected to a gradient shear from 0.01 to 1000 1/s. The conductivity of emulsions was determined with a benchtop conductivity meter at room temperature.

Infrared spectra of nanofibers were analyzed using attenuated total reflectance (ATR) spectroscopy, covering a range of 400–4000 cm^−1^ with a resolution of 4 cm^−1^ over 64 scans. Electrospun nanofiber mat cut into 8 mm squares, mounted on aluminum SEM stubs with conductive carbon tape, then coated with a 5 nm gold layer via sputter deposition prior to morphological examination, and imaged with a JSM-6390LV scanning electron microscope (JEOL Ltd., Tokyo, Japan). To determine the internal structure of different electrospun fibers, electrospinning was initially performed for two minutes, and then a TEM copper grid was adhered onto the electrospinning collector plate to capture fibers for 15 s. After ensuring fiber adhesion using an optical microscope, the fibers were subsequently examined using a JEM-2100 F transmission electron microscopy (Tokyo, Japan).

### 2.6. Microbial Assay

*Lactobacillus plantarum* at a concentration of 1 × 10^9^ CFU/mL was inoculated into sterilized MRS broth culture medium and cultured at 37 °C for 22 h. Subsequently, the bacteria were transferred into three separate 50 mL centrifuge tubes, balanced using tubes containing water, and centrifuged at 8000 rpm for 10 min at 4 °C. The bacteria were washed using phosphate-buffered saline (PBS), thoroughly mixed on a vortex mixer, and centrifuged again. This washing step was repeated three times to completely remove the culture medium, after which the bacteria were suspended in 5 mL ultrapure water and stored at 4 °C until use.

The simulated gastric juice was prepared as described. Briefly, ultrapure water (100 mL) was adjusted to pH 2.5 and sterilized using an autoclave before adding pepsin enzyme (330 mg). The solution was subsequently passed through a Millipore filter (0.22 μm). For simulated intestinal fluid, ultrapure water (100 mL) was mixed with potassium dihydrogen phosphate (1.36 g) and bile salts (330 mg). The pH was adjusted to 6.8 and sterilized, after which precisely 1 g pancreatic enzyme was added. The solution was filtered through a Millipore filter (0.22 μm).

The electrospun fiber loaded with probiotic was cut into 10 mm discs and eluted in PBS (30 min, gentle shaking) to obtain probiotic suspensions. Viability was quantified using the drop plate method. A total of 100 µL of each dilution was dispensed onto quadrant-marked agar plates, and then the colonies were counted after 24 h incubation.

### 2.7. Statistical Analysis

All experiments were performed in triplicate (*n* ≥ 3). Results were displayed as mean ± standard deviation. One-way ANOVA analysis followed by Tukey’s test determined significance, where probability values below 0.05 indicated meaningful differences.

## 3. Results and Discussion

### 3.1. Characterization of W/W Emulsions

Water-in-water (W/W) emulsions are colloidal systems comprising two immiscible aqueous phases formed by the phase separation of hydrophilic macromolecules that are thermodynamically incompatible in solution. The formation of a water-in-water (W/W) emulsion from gelatin (GE) and dextran (DEX) is driven primarily by excluded volume effects resulting from macromolecular crowding and thermodynamic interactions. Spatial repulsion among molecules increases the total free energy of the system, inducing instability and phase separation [29]. Leveraging these thermodynamic instabilities, an aqueous two-phase system composed of gelatin and dextran was successfully emulsified under external force. Previous research has indicated that phase inversion phenomena in W/W emulsions may result from discontinuous changes in the modulus or viscosity of the continuous phase [30]. To investigate the phase distribution within the W/W emulsion systems, dextran was labeled with fluorescein isothiocyanate (FITC). As shown in Figure 1, the green regions correspond to the FITC-labeled dextran-rich phase. Our results demonstrated that at a fixed gelatin concentration of 10 wt%, dextran acted as the continuous phase at lower dextran concentrations (1.5 wt%, 3.0 wt%, 4.5 wt%, 6.0 wt%, and 7.5 wt%). GE and DEX are thermodynamically incompatible; as their overall concentration increases, polymer–polymer repulsion exceeds polymer–water affinity, and the system separates into a DEX-rich phase and a GE-rich phase. Under mechanical agitation, the phase with the smaller volume (dextran) fraction forms droplets (internal phase), while the major phase (gelatin) becomes continuous. When the DEX concentration is low, the GE-rich phase dominates the volume, so DEX droplets are stabilized within the GE matrix. Meanwhile, it is obvious that increasing the dextran concentration (9.0 wt%, 10.5 wt%, 12.0 wt%, 13.5 wt%, and 15.0 wt%) resulted in phase inversion, where dextran transitioned to become the dispersed phase. As the dextran concentration is raised beyond 9 wt%, its volume fraction approaches and then surpasses that of the GE phase. At the point where the two phases are roughly equal in volume, interfacial stresses generated during mixing can no longer be accommodated by the original topology, and the emulsion inverts: the DEX-rich phase becomes continuous, while GE-rich domains become dispersed. This behavior is well-documented for W/W systems in which both phases have ultralow interfacial tension. Increasing the DEX concentration also raises the viscosity of the DEX-rich phase, promoting its dominance as the continuous phase after inversion and stabilizing the new morphology.

To further explore the impact of the DEX concentration on the droplet size in the W/W emulsion system, optical microscopy was employed to observe W/W emulsions prepared with varying dextran concentrations. As illustrated in the Figure 2, the droplet size exhibited an initial increase followed by a subsequent decrease with the rising dextran (DEX) concentration. Specifically, at a constant gelatin (GE) concentration of 10 wt%, dextran acted as the continuous phase at lower concentrations (1.5 wt%, 3.0 wt%, 4.5 wt%, 6.0 wt%, and 7.5 wt%), resulting in an increase in droplet size with the rising DEX concentration. This increase can be attributed to enhanced droplet aggregation and coalescence phenomena due to insufficient intermolecular entanglement within the continuous phase at lower DEX concentrations. However, as the DEX concentration exceeded a critical threshold, causing a phase inversion (at concentrations ≥ 9.0 wt%), the continuous phase transitioned to gelatin (GE), with DEX becoming the dispersed phase. Under these conditions, further increases in DEX concentration (9.0 wt%, 10.5 wt%, 12.0 wt%, 13.5 wt%, and 15.0 wt%) led to stronger intermolecular entanglement within the dispersed DEX-rich phase, restricting droplet coalescence and consequently decreasing the droplet size. These observations underline the dual influence of polymer concentration and phase continuity on droplet morphology, providing practical guidance for optimizing emulsion stability. The turning point at 9 wt% DEX corresponds to a segregated phase inversion in which the volume fraction and viscosity of the dextran-rich phase surpass those of the gelatin-rich phase. Because both phases have ultralow interfacial tension [31], the system minimizes free energy by exchanging roles: gelatin becomes continuous, and dextran becomes dispersed.

Viscosity is an important factor influencing the formation and stability of W/W emulsions, particularly during electrospinning processes [32,33]. Adequate viscosity is critical for successful electrospinning, as insufficient viscosity may lead to unstable fiber jet formation, resulting in electrospraying rather than fiber spinning and ultimately producing particles instead of continuous fibers [34]. Even minor changes in viscosity can significantly impact the structural and morphological properties of the resultant nanofibers. Thus, a detailed investigation into the viscosity behavior of GE/DEX W/W emulsions is essential for optimizing the electrospinning conditions. As shown in Figure 3A, the viscosity tests revealed shear-thinning behavior across all W/W emulsions within a shear rate range of 0.1–1000 s^−1^. The shear-thinning characteristic was consistent across emulsions with various GE and DEX concentration ratios, indicating a universal shear-thinning behavior independent of specific compositions. Notably, viscosity consistently increased with concentration increase, attributed primarily to enhanced intermolecular interactions and entanglements between polymer chains at elevated concentrations. Such entanglements promote stable polymer jet formation during electrospinning, as molecular entanglement is a critical factor influencing the stability of polymer jets and fiber continuity. Consequently, polymer concentration emerges as a decisive parameter affecting the morphological quality of electrospun fibers.

The electrical conductivity of polymer solutions is widely recognized as an important factor influencing the electrospinning process [35]. This is primarily because the formation and stability of the electrospun jet are directly linked to the behavior of charged polymer solutions traveling toward a grounded collector under an applied electric field. Polymer solutions with insufficient conductivity often fail to form stable jets, and the jet breaks into droplets, resulting in compromised spinnability, incomplete fiber formation, or undesired electrospraying, leading to particle formation instead of fibers [36]. Consequently, accurately evaluating and optimizing the electrical conductivity of spinning solutions is essential for achieving stable electrospinning processes and the desired nanofiber morphologies. In order to further explore the influence of polymer concentration on electrical conductivity, W/W emulsions composed of gelatin (GE) and dextran (DEX) at different concentrations were measured. As illustrated in Figure 3B, the electrical conductivity of GE/DEX W/W emulsions increased consistently with increasing DEX concentration. This phenomenon might be due to the intrinsic properties of dextran molecules; a higher polymer content thus directly translates into a higher ion concentration and, consequently, greater electrical conductivity [37]. Specifically, at higher DEX concentrations, the increased availability of charged species or ionic groups, originating from the dextran polymer chains, elevates the solution’s overall ion concentration, thereby increasing its electrical conductivity. The observed relationship between conductivity and DEX concentration further emphasizes the critical role of polymer composition in electrospinning. Lower conductivity often results in weaker jet formation or reduced spinnability, potentially leading to fiber defects or undesired particle formation due to insufficient electrical forces to stabilize and elongate the polymer jet [38]. Conversely, higher conductivity promotes more effective jet stabilization, improving fiber uniformity and morphology by facilitating stable electrostatic stretching. This concentration–conductivity relationship reinforces the compositional control of fiber morphology: insufficient conductivity yields weak, unstable jets and beaded or particulate deposits, whereas higher conductivity provides stronger electrostatic stretching, stabilizes the jet, and enhances fiber uniformity.

### 3.2. Characterization of Electrospun Fibers

To further investigate the effect of polymer concentration on fiber morphology, W/W emulsions with varying GE/DEX ratios were electrospun, and the resultant nanofibers were characterized by scanning electron microscopy (SEM). As shown in Figure 4, SEM images revealed a distinct morphological transformation from irregular fibers at low DEX concentrations to smooth, uniform, and bead-free fibers as the DEX concentration increased. This concentration-dependent transition operates through two coupled mechanisms. First, higher GE/DEX levels markedly elevate viscosity by increasing chain overlap and entanglement, thereby supplying the viscoelastic stress required to keep the electrified jet intact [36]. Second, the same concentration rise boosts ionic strength and electrical conductivity, which enhances the charge transport and electrostatic stretching of the jet. Larger viscosity and higher conductivity together enhanced the stability and continuity of the polymer jet under the applied electric field, reducing jet fragmentation and facilitating the formation of uniform fibers. Conversely, at lower concentrations, insufficient entanglements and lower conductivity weakened the jet’s resistance to the electrostatic stretching force, resulting in fiber instability and irregularities [39].

Controlling the nanofiber diameter is crucial due to its direct influence on the mechanical and functional performance of electrospun materials. Fibers with larger diameters typically exhibit improved mechanical strength and tensile properties, and their greater internal volume can accommodate higher payloads, thus enabling controlled release applications. On the other hand, smaller fibers possess higher surface-to-volume ratios, which increase their interaction efficiency with the surrounding environment, enhancing their applicability in adsorption, catalysis, and biomedical interfaces. Therefore, precise control over the nanofiber diameter is essential for optimizing performance in targeted applications.

The average diameter of electrospun fibers showed a clear dependence on the DEX concentration in the GE/DEX W/W emulsions. Increasing the DEX concentration from 1.5 wt% to 7.5 wt% resulted in a corresponding increase in fiber diameter from approximately 139.01 nm to 263.46 nm. However, a further increase to 9 wt% DEX led to an unexpected reduction in fiber diameter (219.90 nm), followed by a gradual increase at higher concentrations, reaching 418.23 nm at 15 wt%. This non-monotonic behavior is likely attributable to a phase inversion phenomenon occurring at elevated DEX concentrations, where gelatin transitioned to become the continuous phase. This inversion significantly altered the viscosity, molecular entanglement, and conductivity of the emulsions, subsequently impacting the fiber diameter. These findings are consistent with the conductivity measurements and confirm that the fiber morphology results from a complex interplay between solution viscosity, conductivity, and polymer-phase continuity in electrospun W/W emulsions.

In order to verify the core–shell structure of the prepared nanofibers, transmission electron microscopy (TEM) analysis was conducted on fiber electrospun from various W/W emulsion systems. In TEM images, differences in contrast primarily reflect variations in sample density; hence, a clear distinction between core and shell layers is visible due to their density contrast. Typically, core–shell nanofibers exhibit darker cores due to their higher density, while the shell appears lighter because of its relatively lower density. Figure 5 clearly illustrates the formation of core–shell structures in the obtained nanofibers. These TEM observations provided direct visual evidence supporting the presence of distinct core–shell morphologies within the electrospun nanofibers derived from different W/W emulsions.

Comparative spectroscopic characterization was conducted to evaluate the structural integrity of electrospun fibers fabricated via GE/DEX water-in-water emulsion. The FTIR spectra shown in Figure 6 demonstrated that no significant chemical reaction occurred between GE and DEX during the electrospinning process, as evidenced by the absence of new characteristic peaks in the spectra for fibers produced from emulsions of various concentration ratios. Specifically, pure GE exhibited a broad absorption band between 3700 and 3000 cm^−1^, attributed to the stretching vibrations of –NH and –OH groups [30]. Moreover, characteristic peaks for GE appeared at approximately 1646, 1538, and 1447 cm^−1^, corresponding to amide I (primarily C=O stretching and C–N stretching vibrations), amide II (N–H bending and C–N stretching vibrations), and amide III (N–H bending and C–N stretching vibrations), respectively. Similarly, no additional peaks were observed in the spectra of nanofibers prepared from different GE/DEX concentration ratios. However, a notable blue shift in these characteristic bands was observed, likely resulting from electrostatic interactions between the protein-rich GE phase and the polysaccharide-rich DEX phase within the fibers. These shifts further confirm the physical interactions rather than covalent bonding between GE and DEX in the electrospun nanofibers.

### 3.3. Structure of Lactobacillus plantarum-Loaded Electrospun Fibers

Two GE/DEX formulations were selected for probiotic encapsulation—10 wt% GE/7.5 wt% DEX and 10 wt% GE/13.5 wt% DEX. These ratios lie on either side of the phase-inversion boundary yet share a viscosity–conductivity window that yields bead-free jets, allowing us to examine how shell chemistry influences bacterial protection. To further evaluate the encapsulation effectiveness of electrospun fibers derived from different W/W emulsions, fluorescently labeled *Lactobacillus plantarum* were electrospun under optimized parameters. As illustrated in Figure 7A,B, the resultant micrographs clearly demonstrated that probiotics were successfully encapsulated within nanofibers irrespective of emulsion compositions, confirming effective immobilization within the fiber structure. Additional structural confirmation was provided by scanning electron microscopy analysis of probiotic-loaded nanofibers. SEM images (Figure 7C1,C2) revealed that despite probiotic cells being larger than the individual fiber diameters, they were effectively embedded within the fiber matrix. Probiotic cells were observed oriented along the longitudinal axis of the fibers, a phenomenon that can be attributed to the elongational flow and electrostatic alignment within the Taylor cone during the electrospinning process. Initially randomly dispersed bacteria in the polymer solution aligned progressively along the jet streamlines, subsequently solidifying into uniformly distributed cells embedded within the fibers. Collectively, CLSM, fluorescence microscopy, and SEM analyses confirmed that electrospun nanofibers fabricated from W/W emulsions could effectively encapsulate probiotics, achieving uniform and stable bacterial incorporation within the fiber network. These findings underscore the promising application of electrospun core–shell nanofibers as protective matrices for probiotics, facilitating the enhanced stability and uniform distribution essential for effective delivery in functional food products.

### 3.4. Survival Ability of Lactobacillus plantarum Loaded Electrospun Fibers

To evaluate the protective efficacy of probiotic-loaded nanofibers during gastrointestinal digestion, an in vitro digestion test was conducted using simulated gastric fluid (SGF) and intestinal fluids (SIF), followed by probiotic enumeration using the drop plate method. As shown in Figure 8A, free probiotics exposed directly to SGF experienced severe viability loss. Immediately after exposure to the acidic gastric environment, rapid reductions in viable bacterial counts were observed, with complete loss of viability after 60 min. This outcome clearly demonstrates the susceptibility of unprotected *Lactobacillus plantarum* cells to harsh gastric conditions, highlighting the necessity for effective encapsulation strategies to maintain probiotic viability and functionality within the gastrointestinal tract. Encapsulated cells, by contrast, survived the entire 120 min SGF exposure, although their survival depended on shell composition: fibers prepared with 7.5 wt% DEX (gelatin shell) retained ~0.7 log more viable cells than those spun from 13.5 wt% DEX (dextran shell). The superior performance of the GE-shell fibers is attributed to the denser, proteinaceous barrier formed by gelatin, which slows proton diffusion and provides stronger acid resistance. Viability assays confirmed that the two topologies confer different levels of gastric protection: fibers with a GE shell retained ~0.7 log more viable cells after 60 min in simulated gastric fluid than fibers with a DEX shell, while both formulations maintained > 83% survival over the full SGF–SIF sequence. Results indicated no significant differences in probiotic viability between the two emulsion compositions after intestinal fluid exposure. However, following 120 min exposure to simulated gastric fluid, encapsulated probiotics exhibited significantly higher survival rates compared to free cells. Notably, the probiotic viability was substantially influenced by the polymer composition and phase arrangement within the nanofibers. At higher DEX concentrations (13.5 wt%), a marked decrease in viability was observed after 30 min, likely because dextran, acting as the fiber shell, provided relatively weaker protection against gastric acid. Conversely, fibers prepared at lower DEX concentrations (7.5 wt%), with gelatin as the outer shell, demonstrated enhanced protection, presumably due to the gelation properties and barrier effect of gelatin. These findings provide critical insights into how variations in shell composition affect probiotic survival during gastrointestinal digestion, suggesting that gelatin as the shell material confers superior protective capabilities. The results indicate that encapsulating probiotics via the electrospinning of GE/DEX-based W/W emulsions effectively maintains probiotic viability through gastric passage and enables potential controlled release within the intestinal environment. The resultant survival loss of less than 1.1 log CFU/g matches or betters the protection reported for electrosprayed alginate-chitosan microcapsules (2–4 log loss) [40] and electrospun core–shell fibers (2 log loss) [41]. Thus, the GE/DEX fibers deliver gastrointestinal stability on par with the best electrohydrodynamic systems while requiring no organic solvents or extra cross-linking. The established core–shell nanofiber structures thus present promising opportunities for enhancing probiotic stability, controlled release, and efficacy in functional food applications.

Probiotics frequently face a range of challenging environmental conditions during production, storage, and application, among which elevated temperatures are especially detrimental to their viability [42]. To investigate whether encapsulation within electrospun nanofibers prepared from different GE/DEX-based W/W emulsions could effectively protect probiotics under high-temperature conditions, thermal stability experiments were conducted. Probiotic-loaded fibers and free probiotic cells were exposed to temperatures of 65 °C for 30 min and 72 °C for 1 min, followed by enumeration using the drop plate method (Figure 8B). After treatment at 65 °C for 30 min, the viable probiotic counts remained at approximately 8.65 log CFU/g and 8.57 log CFU/g for fibers prepared from emulsions with GE/DEX ratios of 10 wt% GE/7.5 wt% DEX and 10 wt% GE/13.5 wt% DEX, respectively, whereas free probiotic cells exhibited a complete loss of viability. Similarly, following a more severe treatment at 72 °C for 1 min, the probiotic-loaded fibers maintained relatively high viability (8.44 and 8.05 log CFU/g), while the free probiotic population drastically decreased to only 3.90 log CFU/g. These results clearly demonstrate that probiotics encapsulated within electrospun nanofibers exhibited significantly enhanced thermal resistance compared to free cells. The exceptional heat resistance of the encapsulated cells can be rationalized by several, mutually reinforcing factors. Rapid solvent evaporation during spinning produces a tight, low-moisture micro-architecture whose water activity is comparable to that achieved by freeze-drying, thereby suppressing metabolic and hydrolytic reactions [43]. The resulting polymer shell surrounds bacterium, limits heat conduction, and restricts molecular motion, so proteins and membranes experience less thermally induced damage. Additionally, the nanofiber matrix may restrict the mobility of probiotic cells, preventing thermal denaturation and helping maintain structural stability. Encapsulating probiotics via electrospun W/W emulsions offers a promising strategy for maintaining high probiotic viability under high-temperature conditions. These findings highlight the advantage of creating core–shell fibers from water-in-water emulsions, which provide robust thermal shielding without the need for organic solvents or post-spin cross-linking, broadening their applicability in thermally demanding food processes.

Maintaining probiotic viability immediately after encapsulation is critical; however, ensuring long-term viability during storage is even more essential for practical applications. To evaluate the protective effect of electrospun nanofibers prepared from W/W emulsions on the viability of encapsulated probiotics, storage experiments were conducted at both room temperature (25 °C) and refrigeration conditions (4 °C). Viable probiotic counts were periodically measured using the drop plate method. As shown in Figure 8C, electrospun fibers effectively preserved probiotic viability at high levels (>8 log CFU/g) throughout the 5-day storage period, regardless of storage temperature. Although probiotic viability slightly decreased over time, reductions remained within one log CFU/g under both storage conditions, with refrigerated samples exhibiting superior viability retention. Similar short-term studies on electrospun gum Arabic-pullulan [44] and gelatin [45] reported losses of 1.5–2 log over the same period, indicating that the electrospun GE/DEX fibers offer equal or superior protection. The improved probiotic viability observed at 4 °C can be attributed to reduced metabolic activity, allowing the bacteria to enter a dormant state, thereby minimizing viability losses. The mechanism underlying the high probiotic viability achieved by electrospun fibers likely involves the rapid evaporation of solvent during the electrospinning process, creating a dry, stable, and protective fiber matrix that effectively immobilizes probiotics in an anhydrous state. This drying mechanism resembles that observed in freeze drying, where probiotics are stabilized by low moisture content, thereby minimizing metabolic activities and preventing viability loss. Collectively, these findings suggest that electrospun probiotic-loaded fibers prepared from W/W emulsions represent a promising strategy for preserving probiotic viability during storage, offering potential advantages for the development of probiotic-based functional food products.

## 4. Conclusions

In conclusion, electrospun nanofibers were successfully fabricated from gelatin/dextran W/W emulsions and demonstrated effective encapsulation and protection of *Lactobacillus plantarum*. By systematically varying the DEX concentration, the emulsion exhibited phase inversion phenomena, influencing droplet size, viscosity, conductivity, and, subsequently, nanofiber morphology. SEM and TEM analyses confirmed uniform, bead-free fibers with distinct core–shell structures, while FTIR spectroscopy indicated no chemical reactions, but highlighted intermolecular electrostatic interactions within fibers. Encapsulated probiotics showed significantly enhanced thermal stability, maintaining high viability after exposure to elevated temperatures (65 °C and 72 °C), as opposed to free probiotics. Additionally, electrospun fibers effectively protected probiotic viability during short-term storage (25 °C and 4 °C) and gastrointestinal digestion simulations, with gelatin serving as the shell material offering superior protection. Electrospinning of GE/DEX W/W emulsions provides a promising encapsulation strategy, effectively improving probiotic stability, storage longevity, and resistance to harsh environmental stresses, highlighting its considerable potential in probiotic delivery and functional food applications.

## Figures and Tables

**Figure 1 foods-14-01725-f001:**
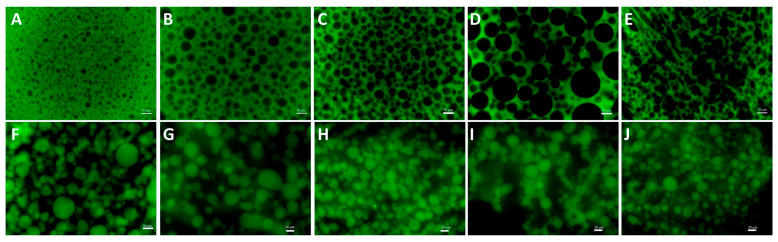
The fluorescence microscope image of GE/DEX W/W emulsions consisting of 10 wt% GE with 1.5 wt% (**A**), 3.0 wt% (**B**), 4.5 wt% (**C**), 6.0 wt% (**D**), 7.5 wt% (**E**), 9.0 wt% (**F**), 10.5 wt% (**G**), 12.0 wt% (**H**), 13.5 wt% (**I**), and 15.0 wt% DEX (**J**), respectively. The green area is rich in FITC-labeled GE; the black area is rich in DEX. The scale bar is 50 μm.

**Figure 2 foods-14-01725-f002:**
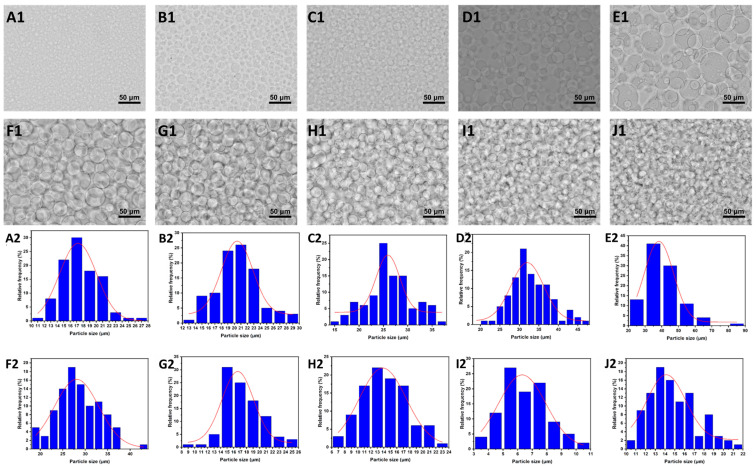
(**A1**–**J1**) The microscope images of GE/DEX W/W emulsions consisting of 10 wt% GE with 1.5 wt% (**A**), 3.0 wt% (**B**), 4.5 wt% (**C**), 6.0 wt% (**D**), 7.5 wt% (**E**), 9.0 wt% (**F**), 10.5 wt% (**G**), 12.0 wt% (**H**), 13.5 wt% (**I**), and 15.0 wt% DEX (**J**), respectively; the scale is 50 μm. (**A2**–**J2**) The diameter distribution of (**A1**–**J1**), respectively.

**Figure 3 foods-14-01725-f003:**
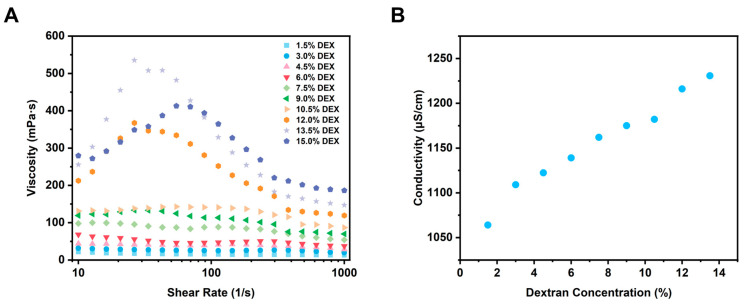
Rheological properties (**A**) and electrical conductivity (**B**) of W/W emulsions at varying dextran concentrations.

**Figure 4 foods-14-01725-f004:**
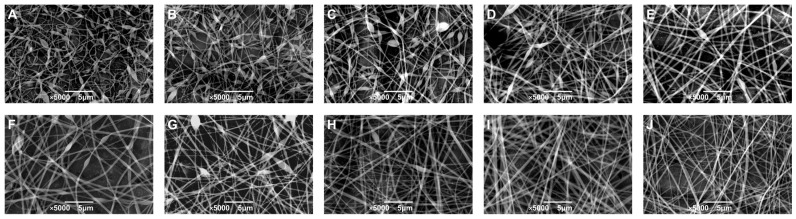
The SEM image of electrospun fibers prepared by W/W emulsions consisting of 10 wt% GE with 1.5 wt% (**A**), 3.0 wt% (**B**), 4.5 wt% (**C**), 6.0 wt% (**D**), 7.5 wt% (**E**), 9.0 wt% (**F**), 10.5 wt% (**G**), 12.0 wt% (**H**), 13.5 wt% (**I**), and 15.0 wt% DEX (**J**), respectively; the scale is 5 μm.

**Figure 5 foods-14-01725-f005:**
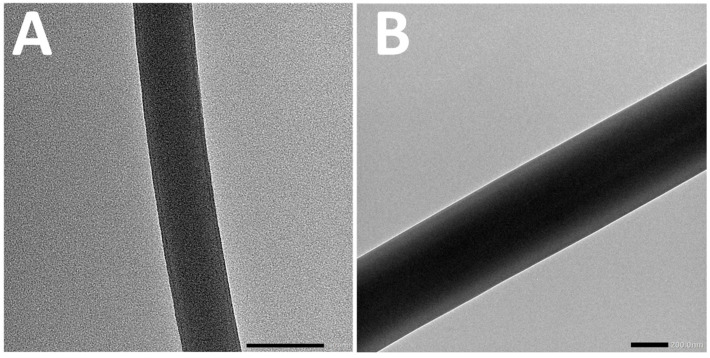
The TEM image of fiber electrospun by 7.5%DEX (**A**) and 13.5%DEX (**B**).

**Figure 6 foods-14-01725-f006:**
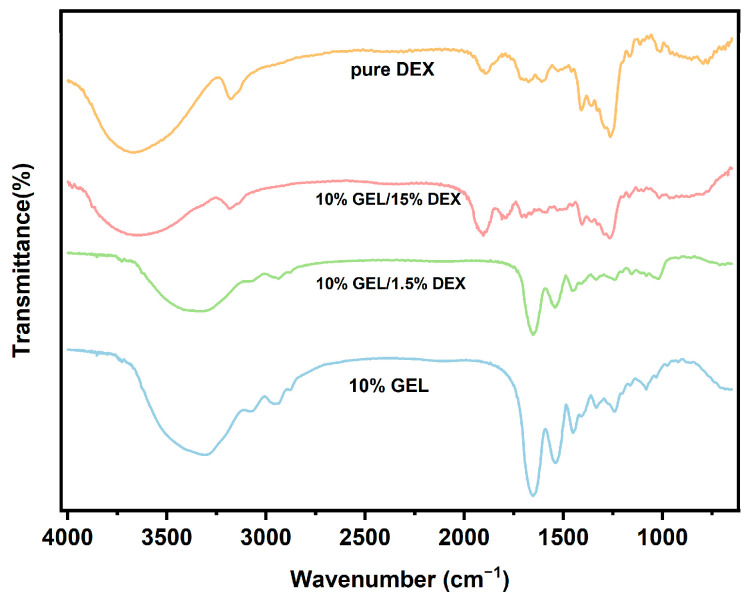
Comparative Fourier-transform infrared spectroscopy analysis of gelatin (GE), dextran (DEX), and their emulsion-based electrospun fibrous composites.

**Figure 7 foods-14-01725-f007:**
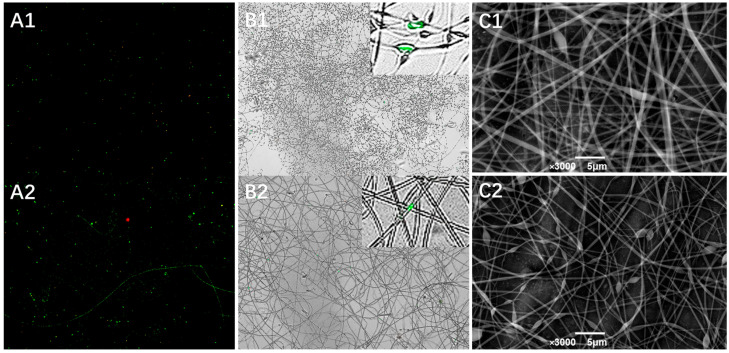
The CLSM image (**A**) and merged image (**B**) of 7.5%DEX (**A1**,**B1**) and 13.5%DEX (**A2**,**B2**) probiotic-loaded electrospun fibers. The SEM image of 7.5%DEX (**C1**) and 13.5%DEX (**C2**) electrospun fiber loaded with *Lactobacillus plantarum*.

**Figure 8 foods-14-01725-f008:**
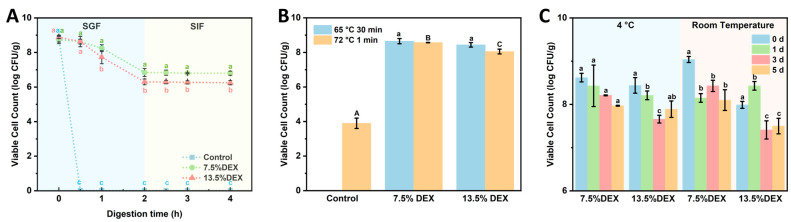
Changes in viable probiotics in 7.5% DEX and 13.5% DEX electrospun fiber after different treatments: (**A**) simulated gastrointestinal digestion, (**B**) heat treatment, (**C**) 5-day storage under 4 °C and room temperature conditions. Distinct lowercase superscripts (a, b, c) and distinct uppercase superscripts (A, B, C) denote statistically significant intergroup variations (*p* < 0.05) determined by one-way ANOVA with Tukey’s post hoc test, with triplicate replicates (*n* = 3).

## Data Availability

The original contributions presented in the study are included in the article, further inquiries can be directed to the corresponding author.

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
