# Peer review of "Electrospun Gelatin/Dextran Nanofibers from W/W Emulsions: Improving Probiotic Stability Under Thermal and Gastrointestinal Stress"

_foods, 2025, doi:10.3390/foods14101725_

Round 1

Reviewer 1 Report

Comments and Suggestions for Authors

The article titled "Electrospun Gelatin-Dextran Nanofibers from W/W Emulsions: Improving Probiotic Stability under Thermal and Gastrointestinal Stress" explores the innovative use of electrospinning technology to enhance the stability of probiotics when exposed to heat and the harsh conditions of the gastrointestinal (GI) tract. However, a few modifications and clarifications are needed to improve the scientific quality of the manuscript.

Avoid writing "we" in the manuscript.

Section 2.2: How the conditions were fixed? Is it by following previously published papers or not? If from a previously published paper, then give proper citation.

Is this electrospinning or electrospraying that the authors used in this work?

Mention the company/manufacturer of the electrospinning machine.

L153-155: Why did the increasing dextran concertation result in phase inversion?

Consequently, polymer 197 concentration emerges as a decisive parameter affecting the morphological quality of electrospun fibers. How?  

The electrical conductivity of polymer solutions is widely recognized as a crucial factor 203 influencing the electrospinning process. Therefore, it is also important to check the zeta potential of the emulsion. However, the same is missing in the manuscript.

L216-218, L222-225: Give proper citation to the fact claimed by the authors.

Figure 8A lacks statical significance (Tukey test)

The results were discussed well; however, they are not verified with previously published results. Therefore, it is advised that the results be verified using recently published research.

Author Response

Thank the reviewer for reviewing our manuscript very much. We express our appreciation for your critical comments and kind suggestions to improve this manuscript. The manuscript has now been revised according to the comments.

Avoid writing "we" in the manuscript.

Author reply: Thanks for your kind suggestions.  All first‑person expressions were converted to passive or neutral forms to maintain an impersonal scientific style.

Section 2.2: How the conditions were fixed? Is it by following previously published papers or not? If from a previously published paper, then give proper citation.

Author reply: In section 2.2, the conditions were not following the previously published papers. They were established through our own preliminary mapping of the dextran/gelatin (DEX–GE) water‑in‑water phase diagram. Concentration pairs spanning 2–20 wt % gelatin and 1–20 wt % dextran was examined, only mixtures that (i) resided inside the stable two‑phase region, (ii) produced uniform droplet morphology after gentle stirring, and (iii) suitable viscosities for continuous fibre formation were retained. The full DEX–GE phase diagram contains >60 data points and is tangential to the central focus of this paper; consequently, we did not include it in the main text. 

Is this electrospinning or electrospraying that the authors used in this work?

Author reply: We confirmed electrospinning: a continuous Taylor cone and fibrous mats (not microspheres) were obtained.

Mention the company/manufacturer of the electrospinning machine.

Author reply: Our electrospinning set‑up is a custom‐assembled bench‑top system. It contains a  DW-9303 high‑voltage source (Dong Wen High Voltage, China) and a LSP01-1A syringe pump (Qili Pump, China). This information has been added in section 2.3.

L153-155: Why did the increasing dextran concertation result in phase inversion?

Author reply: Phase inversion in gelatin/dextran (GE/DEX) water-in-water (W/W) emulsions is governed by the underlying segregative phase separation of two hydrophilic polymers that are mutually repulsive in aqueous media. GE and DEX are thermodynamically incompatible; as their overall concentration increases, polymer–polymer repulsion exceeds polymer–water affinity, and the system separates into a DEX-rich phase and a GE-rich phase. Under mechanical agitation, the phase with the smaller volume fraction forms droplets (internal phase), while the major phase becomes continuous. When DEX concentration is low, the GE-rich phase dominates the volume, so DEX droplets are stabilized within the GE matrix. As DEX is raised beyond ~9 wt %, its volume fraction approaches and then surpasses that of the GE phase. At the point where the two phases are roughly equal in volume, interfacial stresses generated during mixing can no longer be accommodated by the original topology, and the emulsion inverts: the DEX-rich phase becomes continuous, while GE-rich domains become dispersed. This behavior is well-documented for W/W systems in which both phases have ultralow interfacial tension. Increasing DEX concentration also raises the viscosity of the DEX-rich phase, promoting its dominance as the continuous phase after inversion and stabilizing the new morphology. Related information has been added in section 3.1.

Consequently, polymer 197 concentration emerges as a decisive parameter affecting the morphological quality of electrospun fibers. How? 

Author reply: Thank you for requesting clarification. In our system, polymer concentration governs fiber morphology through two coupled effects—viscosity (jet cohesion) and electrical conductivity (charge transport):

Viscosity / chain entanglement – As demonstrated by the rheological data in Figure 3a, solution viscosity rises steadily with increasing GE/DEX concentration. The higher concentration increases molecular overlap and entanglement, providing the viscoelastic stress needed to resist capillary breakup in the electrified jet. At sub-critical concentrations (below the entanglement threshold) insufficient cohesion leads to bead formation; once the threshold is exceeded, smooth, bead-free fibers are produced. We discussed this concentration–morphology relationship in our earlier work (Foods 2023, 12, 1395) and now cite it in Section 3.2.

Electrical conductivity – Concentration also elevates ionic strength, raising the electrical conductivity of the spinning fluid. Higher conductivity enables more efficient charge transport, enhancing jet stretching and further refining fiber uniformity. Conversely, low-conductivity, low-concentration solutions exhibit erratic jetting and produce irregular fibers.

Thus, polymer concentration is “decisive” because it simultaneously dictates the jet’s viscoelastic coherence (via viscosity) and its electrostatic stretching capacity (via conductivity), both of which influence the morphology of electrospun fibers.

The electrical conductivity of polymer solutions is widely recognized as a crucial factor 203 influencing the electrospinning process. Therefore, it is also important to check the zeta potential of the emulsion. However, the same is missing in the manuscript.

Author reply: We did attempt ζ-potential analysis, but the high viscosity and ultralow interfacial tension of these water-in-water emulsions produced unstable, non-reproducible signals. Because reliable ζ-potential values could not be obtained, and because conductivity is the parameter that directly governs jet stability during electrospinning, we characterised charge transport by measuring the solution’s electrical conductivity, which is now reported and discussed in Section 3.1d (Figure 3B).

L216-218, L222-225: Give proper citation to the fact claimed by the authors.

Author reply: We have now added some citations to substantiate the statements in those sentences:

Figure 8A lacks statical significance (Tukey test)

Author reply: We added the statical significance in Figure 8A now.

The results were discussed well; however, they are not verified with previously published results. Therefore, it is advised that the results be verified using recently published research.

Author reply: Thanks for your kind suggestions.  According to your advice, we have now verified our findings against several recently published studies and incorporated these comparisons into the revised Discussion section.

Reviewer 2 Report

Comments and Suggestions for Authors

A brief summary- The manuscript reports the results of the study that aimed to evaluate the protective effect of gelatin-dextran nanofibers on probiotic Lactobacillus plantarum during exposure to gastrointestinal fluids, heat and short-term storage. Authors advocated that encapsulation of probiotics in such structures through electrospinning helped to effectively protect bacterial viability.

• General concept comments- In the opinion of the reviewer, the article requires major revision. The experimental methodology is appropriate, results are presented in a very clear way, and the conclusions are drawn based on them. However, there is very little discussion, and this is a part of the manuscript that requires enhancement.  The authors report as little as 1-2 log reduction of probiotics in response to various stressors using the presented approach, which is much lower than in many other studies where various protective approaches were evaluated and shows that encapsulation of beneficial microbiota in gelatin-dextran nanofibers may be a preferential strategy.

The quality of English throughout the whole text is very high, enabling a good communication of contained science, with very few typographical errors.

• Specific comments

o Title- describes the content of the manuscript well and is sufficiently brief.
o Abstract- The abstract is brief, yet contains some details of results. It is well written.
o Introduction
The introduction conveys background information to the reader in a straightforward way and is very well written. I have only two minor comments for the authors consideration:

The first sentence sounds very familiar. Consider pasting an official definition and placing it in parentheses instead.

Sentence in lines 31-33- this part of background information seems crucial for the content of the manuscript; hence, I would recommend expanding the description of stressors that challenge the survival of probiotics.

o Materials and methods
Section 2.2- Consider placing information about probiotic preparation for electrospinning in front of this section, so the reader is informed about the form the probiotics were added to gelatin-dextran solutions.
Section 2.3- Line 95- 60°C is far from ambient temperature, please explain better what you mean and comment on how this could have impacted the viability of encapsulated probiotics

Section 2.4- seems incomplete, there is no information on what TEM and SEM instruments were used in the study and whether and how the images were analysed. In addition, any sample preparation, such as sputter-coating, should be described.
Section 2.5- Lines 113-115- the word “bacterial” is used here as a typographical error (should say “bacteria”), there are other English mistakes in this section
Section 2.6- It would be appropriate to also state any post-hoc tests for the ANOVA

o Results and discussion

Specific comments:

Caption for Figure 1, 2 and 4 should include information about the dextran concentration in the emulsion shown on each of the micrographs.
Caption for Figure 2- are A1-J1 TEM micrographs? Please specify. These micrographs should have a readable scale bar. Please, consider cropping them to enable increasing the scale bar. How was the size distribution of the droplets measured? This does not seem to be mentioned in the materials and methods section, but it should be there.

Figure 7 should appear below the text that mentions it for the first time.

Section 3.3- Please describe why certain conditions were preferentially selected for L. plantarum encapsulation. A lot of space in this manuscript was devoted to electrospinning and the characterisation of the fibres, which is itself very interesting. However, the key application of these fibres here is the protection of probiotics. Therefore, information that would explain which of gelatin-dextran combinations could be beneficial for encapsulating bacterial cells and why seem crucial.

Section 3.4- The organisation of the text is puzzling at the beginning of this section. The authors first describe the decline of probiotic viability in SGF, but only concerning free cells. Then, the fate of encapsulated probiotics in SIF Is discussed. Consider describing first how all the samples behaved in SGF and then in SIF (even if the viability of free cells was not measured there).

Figure 8- in the caption, there should be a mention of statistics used and wording explaining letters above columns shown in the graphs

Overall comments- the results are very well documented and presented in a suitable, easy-to-follow manner. However, the section completely lacks discussion. It would be of interest tocompare the results of other trials that used electrospinning, electrospraying or other methods for probiotic encapsulation and measured bacterial survival. Experimental conditions and probiotics used will affect these results, but a discussion is still possible.

o Conclusions- are appropriate and drawn based on reported results, but could be expanded if discussion is added to previous section.
o Figures – Are of very good quality, but captions require some additional explanatory notes as suggested in the comments above.
o References- The list contains an appropriate number of references as per a research paper(36). Nevertheless, the manuscript contains little discussion. Therefore, more references are required. All of the references presented in the current version of the manuscript are justified, and a great majority is recent (<5 years).

Author Response

Thank the reviewer for reviewing our manuscript very much. We express our appreciation for your critical comments and kind suggestions to improve this manuscript. The manuscript has now been revised according to the comments.

  • Specific comments

o Title- describes the content of the manuscript well and is sufficiently brief.

Author reply: Title- describes the content of the manuscript well and is sufficiently brief.

o Abstract- The abstract is brief, yet contains some details of results. It is well written.

Author reply: Thank you for your positive feedback on the abstract.

o Introduction

The introduction conveys background information to the reader in a straightforward way and is very well written. I have only two minor comments for the authors consideration:

The first sentence sounds very familiar. Consider pasting an official definition and placing it in parentheses instead.

Author reply: Thank you for this suggestion. We have revised the opening sentence of the Introduction to incorporate the official FAO/WHO definition of probiotics.

Sentence in lines 31-33- this part of background information seems crucial for the content of the manuscript; hence, I would recommend expanding the description of stressors that challenge the survival of probiotics.

Author reply: We have expanded the relevant passage to give a clearer overview of the main environmental challenges faced by probiotic cells during processing, storage, and gastrointestinal transit. We believe the expanded description provides the necessary context for the encapsulation strategy presented in our study.

o Materials and methods

Section 2.2- Consider placing information about probiotic preparation for electrospinning in front of this section, so the reader is informed about the form the probiotics were added to gelatin-dextran solutions.

Author reply: Thank you for this helpful suggestion. We have located the probiotic‐preparation paragraph to Section 2.2.

Section 2.3- Line 95- 60°C is far from ambient temperature, please explain better what you mean and comment on how this could have impacted the viability of encapsulated probiotics

Author reply: Thank you for highlighting this point. Gelatin rapidly gels at room temperature, causing the GE/DEX emulsion to solidify and block the needle. Preliminary trials showed that maintaining the spinning chamber at ≈ 60 °C prevents gelatin gelation and keeps the emulsion fluid enough for continuous fiber formation. To verify that this elevated temperature did not compromise L. plantarum, we enumerated viable cells before and after 30 min exposure to the 60 °C environment and observed no significant reduction

Section 2.4- seems incomplete, there is no information on what TEM and SEM instruments were used in the study and whether and how the images were analysed. In addition, any sample preparation, such as sputter-coating, should be described.

Author reply: We have expanded Section 2.4 to provide full details of instrumentation, sample preparation, and image processing. And the sample were prepared by sputter-coating method.

Section 2.5- Lines 113-115- the word “bacterial” is used here as a typographical error (should say “bacteria”), there are other English mistakes in this section

Author reply: The wording has been corrected and the full subsection re-edited by a native speaker.

Section 2.6- It would be appropriate to also state any post-hoc tests for the ANOVA

Author reply: We now state that Tukey’s HSD was applied after one-way ANOVA.

o Results and discussion

Specific comments:

Caption for Figure 1, 2 and 4 should include information about the dextran concentration in the emulsion shown on each of the micrographs.

Author reply: Thank you for pointing this out. We have updated the captions for Figures 1, 2, and 4 to specify the dextran (DEX) concentration corresponding to each micrograph.

Caption for Figure 2- are A1-J1 TEM micrographs? Please specify. These micrographs should have a readable scale bar. Please, consider cropping them to enable increasing the scale bar.

Author reply: Thank you for pointing this out. Panels A1–J1 in Figure 2 are indeed optical microscope image. We have cropped each image to enlarge key structural details; and inserted a readable scale bar in every panel.

How was the size distribution of the droplets measured? This does not seem to be mentioned in the materials and methods section, but it should be there.

Author reply: We apologies for the omission. The droplet-size distribution was obtained from optical micrographs using ImageJ software. And the description has been added to Section 2.5.

Figure 7 should appear below the text that mentions it for the first time.

Author reply: Thank you for noting the placement issue. Figure 7 has been repositioned so that it now appears immediately after the paragraph in which it is first cited, ensuring smooth flow between text and figure.

Section 3.3- Please describe why certain conditions were preferentially selected for L. plantarum encapsulation. A lot of space in this manuscript was devoted to electrospinning and the characterisation of the fibres, which is itself very interesting. However, the key application of these fibres here is the protection of probiotics. Therefore, information that would explain which of gelatin-dextran combinations could be beneficial for encapsulating bacterial cells and why seem crucial.

Author reply: Thank you for highlighting this point. We have added a clarifying paragraph to Section 3.3 that explains the rationale for choosing the 10 wt % GE/7.5 wt % DEX and 10 wt % GE/13.5 wt % DEX formulations for probiotic encapsulation. We selected two specific GE/DEX formulations because they bracket the phase-inversion boundary and, at the same time, lie within the viscosity and conductivity range that produced stable, bead-free jets.  At 7.5 wt % DEX electrospinning produces fibers with a hydrophilic DEX-rich shell surrounding a GE-rich core, favouring rapid probiotic release in intestinal fluid. At 13.5 wt % DEX the mixture has just crossed the inversion threshold, giving fibers whose protein-based GE shell forms a tighter, acid-resistant barrier that slows proton diffusion. Comparative viability tests show that GE-shell fibers retain about 0.7 log more L. plantarum after 60 min in simulated gastric fluid than the DEX-shell fibers, while both formulations provide > 83 % survival after the full SGF–SIF sequence. Thus, these two gelatin–dextran combinations were deliberately chosen to illustrate how phase topology and resulting shell chemistry directly influence probiotic protection in harsh gastrointestinal conditions. Also, the related information has been added in the revised discussion section.

Section 3.4- The organisation of the text is puzzling at the beginning of this section. The authors first describe the decline of probiotic viability in SGF, but only concerning free cells. Then, the fate of encapsulated probiotics in SIF Is discussed. Consider describing first how all the samples behaved in SGF and then in SIF (even if the viability of free cells was not measured there).

Author reply:  We appreciate this suggestion and have completely reorganized the opening of Section 3.4 to follow the requested order: (1) Survival in simulated gastric fluid (SGF) is now discussed first for both free and encapsulated cells in a single paragraph. (2)Survival in simulated intestinal fluid (SIF) is then presented for the same sample set in the next paragraph, allowing a clear, sequential comparison.

Figure 8- in the caption, there should be a mention of statistics used and wording explaining letters above columns shown in the graphs

Author reply: Thank you for pointing this out. The caption for Figure 8 has been updated to include the statistical test employed and an explanation of the lettering scheme.

Overall comments- the results are very well documented and presented in a suitable, easy-to-follow manner. However, the section completely lacks discussion. It would be of interest to compare the results of other trials that used electrospinning, electrospraying or other methods for probiotic encapsulation and measured bacterial survival. Experimental conditions and probiotics used will affect these results, but a discussion is still possible.

Author reply: Thank you for the valuable suggestions. We have revised the manuscript to address every specific comment you provided, and, in addition, we have substantially expanded the Discussion section to place our results in a broader scientific context.

o Conclusions- are appropriate and drawn based on reported results, but could be expanded if discussion is added to previous section.

Author reply: Thank you for this recommendation. Because we have substantially expanded the Discussion to compare our data with recent literature, we have also lengthened the Conclusions to capture those additional insights.

o Figures – Are of very good quality, but captions require some additional explanatory notes as suggested in the comments above.

Author reply: Thank you for the helpful suggestion. We have revised every figure caption to include suitable additional explanation.

o References- The list contains an appropriate number of references as per a research paper(36). Nevertheless, the manuscript contains little discussion. Therefore, more references are required. All of the references presented in the current version of the manuscript are justified, and a great majority is recent (<5 years).

Author reply: Thank you for pointing this out. In conjunction with the expanded Discussion, we have added several new, recent references that benchmark our findings against other electrospinning-, electrospraying-, and spray-drying-based probiotic systems. More than 70 % of the citations now published within the past five years.

Round 2

Reviewer 2 Report

Comments and Suggestions for Authors

The authors made substantial changes to the body of the manuscript and addressed most comments adequately, especially by adding some discussion that was lacking in the previous version. Therefore, I recommend article acceptance. 

Comments on the Quality of English Language

English is of good quality, and only minor errors that do not impede the understanding of the text must be addressed during the further editorial process.